# Personal Resources and Spiritual Change among Participants' Hostilities in Ukraine: The Mediating Role of Posttraumatic Stress Disorder and Turn to Religion

Iwona Niewiadomska [1], Krzysztof Jurek [2,*], Joanna Chwaszcz [1], Patrycja Wośko [1] and Magdalena Korżyńska-Piętas [3]

[1]  Department of Social Psychoprevention, John Paul II Catholic University of Lublin, 20-950 Lublin, Poland; iwona.niewiadomska@kul.pl (I.N.); chwaszcz@kul.pl (J.C.); patrycja.wosko@gmail.com (P.W.)

[2]  Department of Sociology of Culture, Religion and Migration, John Paul II Catholic University of Lublin, 20-950 Lublin, Poland

[3]  Chair and Department of Development in Midwifery, Medical University of Lublin, 20-059 Lublin, Poland; korzynska.magdalena@gmail.com

*  Correspondence: kjurek@interia.eu

**Abstract:** The theory of conservation of resources (COR) can be used for searching mechanisms which explain spiritual changes caused by trauma. The aim of this paper was to analyze the relationship between distribution of personal resources and spiritual change, as well as the mediating role of posttraumatic stress disorder (PTSD) and turn to religion (stress coping strategy) in this relationship among participants' hostilities in Ukraine. A total of 314 adults—74 women and 235 men—participated in the study. The mean age was 72.59. Polish adaptation of Hobfoll's Conservation of Resources-Evaluation (COR-E), the Posttraumatic Stress Disorder (PTSD) Checklist—Civilian Version (PCL-C), the Inventory for Measuring Coping with Stress (MINI-COPE), and The Posttraumatic Growth Inventory (PTGI) were employed in the research. The mediating role of posttraumatic stress disorder and turn to religion in relationship between personal resources loss and spiritual change was confirmed. The turn to religion plays the role of mediator in relationship between personal resources gain/assigning value to personal resources and spiritual change. The results justify the postulate of conducting further research in the field of testing models which take into account the relationship between posttraumatic stress disorder, religious coping stress, and posttraumatic spiritual change. The conducted analyses should include the assumptions of the COR theory as well as psychological, social, and situational factors that could generate spiritual change.

**Keywords:** personal resources; spiritual change; posttraumatic stress disorder; turn to religion

## 1. Introduction

Posttraumatic stress disorder (PTSD) is described as a delayed and/or prolonged reaction to highly stressful events which are associated with loss of life, serious bodily harm, and/or danger to the physical integrity of oneself, accompanied by intense anxiety, feeling of helplessness and/or danger. PTSD is characterized by symptoms such as: recurring images, dreams, thoughts related to the experienced event (intrusion), attempts to avoid thinking about the situation and its experiences (avoidance), manifesting aroused vigilance, fear, difficulty concentrating, and/or overstimulation (Kessler et al. 1995; Ogińska-Bulik and Juczyński 2012; Ogińska-Bulik 2016). The detailed PTSD criteria are included in The International Classification of Diseases and Related Health Problems ICD-11, Diagnostic and Statistical Manual of Mental Disorders V (DSM-V). Hostilities are one of the biggest stressors and often lead to serious mental health problems. Warfare can be broadly understood as any action taken during a war to achieve its goals. In this perspective, these are both military and non-military actions. The division into soldiers and civilians is blurred. The main

field of contemporary warfare (e.g., by Russia in Ukraine) is not physical space, but above all the unlimited sphere of psychological influence. These actions are not aimed at physically annihilating the enemy and seizing his territory, but their aim is to break the enemy's will to fight and limit his possibilities of resistance (Jonsson and Seely 2015). While the severity of the problem of PTSD in people who participate in hostilities is unambiguous, the results of research are inconsistent. For example, in studies conducted by Hoge et al. (2004), the syndrome was diagnosed in 18% of Iraqi and 11.5% of Afghanistan war veterans. Hall et al. (2008) reported the occurrence of the syndrome in 26% of Gaza settlers. Groth et al. (2013) found PTSD in 56% of the surveyed soldiers who participated in peacekeeping missions in Iraq or Afghanistan. Among adult immigrants and refugees who were fleeing the war, the incidence of PTSD was ranged from 2.2% to 88.3% (Bogic et al. 2015; Morina et al. 2018). A traumatic experience can lead to different effects. There are three possible consequences of experiencing trauma: (1) survival, which is described as a lower level of functioning compared to the period before the trauma; (2) balance recovery; (3) thriving and growth, which is characterized by an increase in the level of efficiency compared to the period before the trauma (O'Leary et al. 1996). Spiritual change is one of the five dimensions of posttraumatic growth. Its essence includes, among other things, changes in the philosophy of life in the form of increased affirmation and meaning in life, greater interest in the religious dimension in the form of strengthening the religious attitude, experiencing greater joy from everyday/small events. In addition to spiritual changes, posttraumatic growth includes positive changes in self-perception (towards perceiving oneself as better, stronger, more valuable, more effective, more confident) and constructive changes in interpersonal relationships (towards greater sensitivity, compassion, and unbosoming oneself to others). The most positive changes in the field of posttraumatic growth occur in the period from two weeks to two months after the experienced trauma. However, it was also noticed that the described changes may occur many years after the traumatic event (Tedeschi and Calhoun 1996, 2004; Nishi et al. 2010; Ogińska-Bulik and Juczyński 2012; Ogińska-Bulik 2016). In the literature, there is no unanimous position on the relationship between the magnitude of the trauma and the severity of posttraumatic transformations in the spiritual sphere. Some authors claim that the size of posttraumatic changes depends on the intensity of the trauma, i.e., the more intense the traumatic experience, the greater the chance of positive changes as a result of this type of experience (McMillen et al. 1997; Park et al. 1996). In contrast, other authors indicate that this is a curvilinear relationship, i.e., the trauma of medium intensity favors transformation in the way of perceiving circumstances, while events of low or extremely high intensity do not lead to such changes (Linley and Joseph 2004).

Spiritual change in the context of the experienced trauma refers to a better understanding of spiritual problems and/or a deepening of the spiritual experience. The presented understanding of changes in the spiritual sphere means that before the occurrence of the trauma, the person had spiritual experiences (in such circumstances, traumatic events may lead to their intensification or decline), or one of the effects of the trauma is a shift towards triggering spiritual experiences. The change in the spiritual dimension is the result of the transformation which takes place in the way of perceiving the events that took place by processing the information contained in the cognitive scripts on the experienced trauma in the context of searching for the meaning of the event and/or the emergence of reflection on its significance for further functioning. Posttraumatic changes in the spiritual sphere are more than a return to a state of equilibrium after an experienced traumatic experience. They are the result of searching for the meaning of experienced suffering. Moreover, posttraumatic changes do not exclude the occurrence of psychological discomfort and/or lowering the quality of life as a result of the stress coping strategies which were used. In this article the stress coping strategy known as turning to religion is analyzed. The strategy should be understood as an active coping with stress, the essence of which is the fact that religion can become a source of emotional support, a way-indicator for a positive revaluation.

### 1.1. Relationship between Conservation of Resources, Posttraumatic Stress Disorder, and Spiritual Change

The theory of conservation of resources (COR) was used for searching mechanisms which explain spiritual change. In the last 30 years the theory has become one of the most frequently used approaches in psychology which explains the mechanisms regulating human behavior, including explaining PTSD (Hobfoll 2001; Hobfoll 2002; Hobfoll et al. 2006a; Chen et al. 2015; Hobfoll et al. 2018). The COR theory assumes that stress occurs when resources which are essential for functioning have been lost or there is a risk of their loss (Heath et al. 2012; Hollifield et al. 2016; Hobfoll et al. 2018). Particularly important predictors of PTSD are losses of entire groups of resources related to each other in teams/caravans—including loss of personal resources in the form of a caravan, which is constituted by a sense of security and attachment, hope for the future, perception of self-worth, self-efficacy, and a sense of bond with relatives (Hobfoll et al. 2007; Hobfoll 2010; Hobfoll et al. 2020). The resources losses do not have to occur during the experienced trauma. Most frequently they are the effect of the long-term loss of resource process before and/or after a traumatic event, which is described as long-term risk cascades of resource losses (King et al. 1999; Vogt et al. 2011; Interian et al. 2014). In some studies, a significant relationship between perception of the resources and experiencing PTSD was found. Moreover, the symptoms of PTSD in the situation of resource gains appeared in a smaller severity than in the situation of experiencing losses (Slobodin et al. 2011; Hall et al. 2014). According to S. Hobfoll's theory of resource distribution (COR), functioning in traumatic circumstances is associated with a high intensity of resource distribution, mainly with multidimensional losses, and less frequently with the perception of resource gains. The categories of resource distribution, in turn, indicate posttraumatic changes. Experienced losses (especially long-term loss cycles) increase the risk of negative effects (e.g., depression), and perceived gains—a higher probability of experiencing development/posttraumatic growth in various areas of life (Frazier et al. 2001; Sattler et al. 2002; Ai et al. 2005; Hobfoll et al. 2006b, 2007; Hall et al. 2008; Kaniasty and Norris 2008; Palmieri et al. 2008; Hobfoll 2010; Galatzer-Levy and Bonanno 2014; Morina et al. 2018; Pietrzyk et al. 2017; Prucnal et al. 2017; Hobfoll et al. 2018, 2020). However, the studies which do not confirm above relationships are presented in the literature (Tomich and Helgeson 2004; Hobfoll et al. 2006a; Zoellner and Maercker 2006; Hobfoll et al. 2011; Kaniasty 2012; Wośko et al. 2017).

### 1.2. Relationship between Posttraumatic Stress Disorder and Spiritual Change

In the literature, three models of the relationship between PTSD and spiritual change can be indicated. The first model assumes that PTSD and spiritual change are the opposite ends of the continuum. Consequently, there is observed a negative relationship—a lower severity of stress symptoms generates higher spiritual change (Taylor et al. 2000; Frazier et al. 2001; Zoellner and Maercker 2006). According to the second model, it is assumed that spiritual change coexists with PTSD in a positive way. The higher the level of posttraumatic stress, the greater the intensity of spiritual change. The changes are generated by stress in the cognitive sphere (constructive transformation of cognitive structures related to trauma) and in the emotional sphere (making efforts motivated by tension and anxiety) towards making sense of the traumatic event and its religious, philosophical, and/or existential consequences (Tedeschi and Calhoun 2004; Solomon and Dekel 2007; Dekel et al. 2012). In this approach, for the occurrence of posttraumatic spiritual change, four elements are necessary: (1) a serious trauma (not mild stress); (2) significant life changes; (3) experiencing growth as a result of coping with trauma, and not a strategy/mechanism of coping with strong stress; (4) radical changes in the assumptions/philosophy of life. Moreover, these changes should not be treated as an element of the individual's adaptation, because they may be accompanied by a sense of mental discomfort, a reduction in the sense of quality of life, some symptoms of PTSD (e.g., intrusion and/or avoiding traumatic thoughts), or confronting the profits and losses resulting from the trauma (Tedeschi and Calhoun 1996, 2004; Helgeson et al. 2006; Hobfoll et al. 2006b; Ogińska-Bulik and

Juczyński 2010). The third model assumes that spiritual change is independent of PTSD symptoms (e.g., Joseph and Linley 2006; Lepore and Ravenson 2006). From this perspective, spiritual change is not the effect of the experienced trauma, but of the coping strategies undertaken as a result of it (Tedeschi and Calhoun 2004). This approach is supported by the results of research, according to which both people with resilience to PTSD, as well as individuals experiencing even severe trauma symptoms are able to function well in various aspects of life, especially in terms of maintaining personal safety and social relationships (Hobfoll et al. 2011; Kaniasty 2012). According to Israeli research, 23% of previously imprisoned soldiers showed symptoms of PTSD and at the same time declared positive changes in all areas of posttraumatic growth, including growth in the spiritual dimension (Solomon and Dekel 2007). In Polish studies, it was found that only 14% of soldiers participating in peacekeeping missions experienced a high level of posttraumatic growth in all its constituent areas (intrapsychic, interpersonal, and spiritual), but as many as 61% stated that there were constructive changes in the spiritual sphere, especially in terms of higher appreciation for life (Ogińska-Bulik 2016).

### 1.3. Relationship between Religious Coping Strategies and Spiritual Change

Spiritual change is also associated with manifesting activity in the religious sphere. Most research results indicate that there is a positive correlation with mental well-being (greater sense of quality of life and health) with the manifestation of internal religiosity, shown, e.g., by manifesting trust in God, religious practices based on trust, perceiving faith as a source of strength, relying on God's help (Koenig 2012; García-Alandete and Bernabé-Valero 2013; Krause and Hayward 2013; Krok 2014; Dowson and Miner 2015; Zarzycka et al. 2017; Aghababaei et al. 2018; Zarzycka and Puchalska-Wasyl 2020; Thomas and Barbato 2020). In the literature, the opinion that the relationship between religiosity and mental health is a linear one, dominates. However, Galen and Kloet (2011) showed a curvilinear relationship. People with higher belief certainty (both confidently religious and atheists) have greater wellbeing relative to those with low certainty (unsure and agnostics). Experiencing stress is conducive to taking actions in the religious sphere in the form of undertaking religious coping strategies. For example, in March 2020 a significant increase of religious coping was noticed and related to the COVID-19 pandemic. The presented dependence was based on the analysis of the frequency of Google searches of religious terms in 95 countries. On this basis, the highest frequency of searches for the term "prayer" was noted since Google introduced the registration of the frequency of searches of various types of concepts/terms. An increase in searches for terms such as God, Allah, the Bible, the Koran, and the Internet church was also observed at a significantly higher level (Thomas and Barbato 2020). Religious coping with experienced trauma can be helpful (positive), harmful (negative), or neutral (Pargament et al. 2000). For example, in veterans who experienced traumatic events during hostilities, no unequivocal relationships could be found in terms of PTSD symptoms and religious behavior (Tran et al. 2012). Some studies justify the conclusion that experiencing posttraumatic stress favors the weakening of religious behavior (Witvliet et al. 2004), while other analyses—their strengthening (Fontana and Rosenheck 2004). In the context of spiritual change, attention should be paid to the importance of a positive religious strategy to deal with aggravating events. The use of the method of coping reflects human attempts to find sense in confrontation with trauma (Stein et al. 2009). This results from the fact that elements of a religious nature by providing people with an interpretation of painful events, simultaneously help to develop a relatively coherent and meaningful image of reality in which even traumatic events have their sense (Park 2013; Krok 2014; Zarzycka et al. 2020). The results of the meta-analysis conducted on 49 studies support the conclusion that there is a moderate positive association between a positive religious stress coping strategy and the positive effects of experienced stress (Ano and Vasconcelles 2005). The people who used positive religious coping strategies—in the form of, e.g., positive religious reevaluation, seeking spiritual support, attending religious meetings, undertaking religious practices, praying—

experienced more constructive behaviors (growth in the area of spiritual, self-esteem, social relationships, quality of life), and at the same time fewer symptoms of disorders (anxiety, depression) (Solomon and Mikulincer 2006; Tedeschi and Calhoun 2004; Hinton and Kirmayer 2013). However, the conducted analyses do not allow for an unequivocal statement of a causal relationship between the preference for positive religious coping strategies and posttraumatic growth in the spiritual, intrapsychic, and/or interpersonal dimensions (Harrison et al. 2001; Smith et al. 2003; Pargament et al. 2000; Tran et al. 2012; Ellison et al. 2013; Park 2013). It turns out that not all war experiences lead to focusing on a relationship with God. For example, soldiers might use religious and non-religious coping strategies and it may relate to the extent they believed they were able to control the outcome of their combat experience. This experience can be positive or negative (Wansink and Wansink 2013). Moreover, religious coping is not an independent factor in generating constructive behavior. Its action depends on the connection/interaction with variables of a personal, situational, and socio-cultural nature (Ahles et al. 2016; Büssing et al. 2005; Santos et al. 2017; Areba et al. 2018; Bradshaw and Kent 2018; Pirutinsky et al. 2020; Tedeschi and Calhoun 1996, 2004; de Jong 2004). The results presented in the article represent a voice/report in the ongoing discussion in the literature on the mechanisms generating positive spiritual changes resulting from the experienced trauma.

Based on above assumptions, we hypothesized:

**Hypothesis 1.** *PTSD and turn to religion are mediators in the relationship between personal resources gain and spiritual change.*

**Hypothesis 2.** *PTSD and turn to religion are mediators in the relationship between personal resources loss and spiritual change.*

**Hypothesis 3.** *PTSD and turn to religion are mediators in the relationship between assigning value to personal resources and spiritual change.*

## 2. Materials and Methods

### 2.1. Participants and Procedure

The study included 314 adults, 74 women and 235 men (five respondents did not indicate their gender), aged between 18 and 74 years. The mean age was 34.08 years (SD = 9.83). Most respondents lived with a partner ($n$ = 173, 55.1%). In the housing situation of the respondents, two categories dominated: respondents who had their own flat/house ($n$ = 139, 44.3%) and those who rented a house/flat ($n$ = 53; 16.9%). The highest number of respondents had a higher education ($n$ = 134, 42.7%). With regard to the financial situation, the highest number of participants described it as bad or rather bad ($n$ = 144, 46.5%). Most of the respondents were Orthodox ($n$ = 241; 76.8%). Most of the respondents declared that they were religious (definitely or rather) ($n$ = 225; 71.7%). Participants were invited to complete a set of questionnaires and then return them personally to research assistants. No time limitations were imposed on the participants. The participants were fully debriefed of the aim of study, and their queries were explained by the researcher. The study was anonymous.

The study was conducted in 2019 among Ukrainian civilians living in towns located in the Donbass—an area of eastern Ukraine, where an armed conflict has been going on since 2014 between pro-Russian separatists and the Russian Federation supporting them, and the army representing the legal authorities of Ukraine. Displaced people from Donbass, who were temporarily in the central and western parts of Ukraine, also participated in the research. Responses were provided in the presence of trained interviewers of Ukrainian nationality. The supervisors of the study were the employees of the Institute of Psychology and the Institute of Sociological Sciences of the John Paul II Catholic University of Lublin.

The procedure was approved by the Research Ethics Committee at the Institute of Sociological Sciences of the John Paul II Catholic University of Lublin.

*2.2. Measures*

2.2.1. Conservation of Resources (COR) Evaluation

The dynamics of resource conservation were measured using the Polish adaptation of Stevan E. Hobfoll's Conservation of Resources evaluation questionnaire (COR evaluation), prepared by Chwaszcz et al. (2019). The COR evaluation is based on Hobfoll's resource conservation theory, which describes stress as a phenomenon affecting resource management, creating a risk of resource loss, and causing their actual loss, or inhibiting their growth. The questionnaire contains a list of 74 resources. In the first step, the respondents rated each resource on a five-point scale where 1—means not at all, and 5—to a very large extent, in two categories: loss and gain (To what extent have I gained these resources in my life?/To what extent have I lost these resources in my life?). In the second step, the respondents assigned value to each resource where 1—means not value, and 5—a very large value how important are the following resources to me?). Various survey resources were taken into account, e.g., "Family stability", "Good relations with my children", "My children's health", "A sense of closeness with my spouse or partner", "Awareness of the goal to which I am going in life", "Belonging to an organization where I can share my interests with others".

2.2.2. Posttraumatic Stress Disorder (PTSD) CheckList (PCL-C)

The PTSD Checklist (Weathers et al. 1993) is a 17-item scale originally based on the DSM-III-R posttraumatic stress disorder criteria and revised in 1994 to correspond to the DSM-IV criteria. Using a five-point scale (0—means not at all; 4—means often), respondents indicate how much they were bothered by each PTSD symptom in the past month. Initial psychometric data was derived by using a military version of the PCL (PCL-M) in a sample of Vietnam veterans. Alfa Cronbach for the whole scale was 0.97. In the study, the authors used a civil version to enable respondents to report symptoms of increased posttraumatic stress related to any traumatic events, not just symptoms caused by military experiments. Assessment of the symptoms of posttraumatic stress intensification taking into account military and non-military sources of trauma is especially important for considering the general mental health of participants in military operations. Examples of PCL-C items: "Feeling very upset when something reminded you of a stressful experience from the past?", "Avoid thinking about or talking about a stressful experience from the past or avoid having feelings related to it?", "Feeling as if your future will somehow be cut short?".

2.2.3. Coping Inventory (MINI-COPE)

The Polish version of the Mini-COPE (Juczyński and Ogińska-Bulik 2009) inventory was used to measure coping strategies. It is a shortened version of the Multimodal Inventory for Measurement of Coping with Stress-COPE (by Carver, Scheier, and Weintraub) and measures coping in terms of disposition. It consists of 28 statements that are part of 14 strategies for coping with stress, including active coping, planning, positive revalidation, acceptance, sense of humor, turn to religion, seeking emotional support, seeking instrumental support, taking care of something else, denial, discharge, use of psychoactive substances, cessation of activities, and self-blaming. There are two theorems for each strategy. The tested respondent refers to each statement by marking one possible answer on a four-point scale where 0—means "I almost never do so" and 3—means "I almost always do so". The obtained psychometric properties are satisfactory. The half-reliability for 14 scales is 0.86 (Guttman's index 0.87). In this study, we analyzed only one stress coping strategy—turn to religion. The domain consists of two items "I've been trying to find comfort in my religion or spiritual beliefs" and "I've been praying or meditating".

2.2.4. Posttraumatic Growth Inventory (PTGI)

The Posttraumatic Growth Inventory (PTGI) (Tedeschi and Calhoun 1996) was used to measure the degree of positive change experienced in the aftermath of the loss that the participants identified as the most traumatic in the past five years. Each item was rated

using a six-point scale where 0—means "I did not experience this change as a result of my crisis" and 5—means "I experienced this change to a very great degree as a result of my crisis". The Alpha Cronbach index for the total 21-item PTGI is 0.90, while for individual domains it ranged from 0.67 to 0.85. The PTGI consists five domains, i.e., relating to others, new possibilities, personal strength, spiritual change, and appreciation of life. In this study, we analyzed only one domain—spiritual change. The domain consists of two items: "A better understanding of spiritual matters" and "I have a stronger religious faith".

### 2.3. Statistical Methods

Descriptive statistics (means, standard deviations, and alpha reliabilities) were calculated for variables. The correlations between personal resources (loss, gain, assigning value to personal resources), PTSD, turn to religion, and spiritual change were conducted. Finally, we examined whether the relationships between personal resources and spiritual change are mediated by PTSD and turning to religion. Three separate mediation models were tested to assess direct and indirect effects of personal resources loss, personal resources gain, and assigning value to personal resources on spiritual change mediated through PTSD and turn to religion (Figure 1). We performed analysis using SPSS's add-on Process. Indirect effects and bias-corrected confidence intervals (CI 95%) were calculated using the bootstrapping procedure (5000 bootstrapped samples). If this CI does not include zero, the analyst concludes that there is statistically significant mediation. The completely standardized indirect effect ($ab_{cs}$) was employed as an estimate of effect size (0.01 = small effect size; 0.09 = moderate effect size; and 0.25 = large effect size (Preacher and Kelley 2011)).

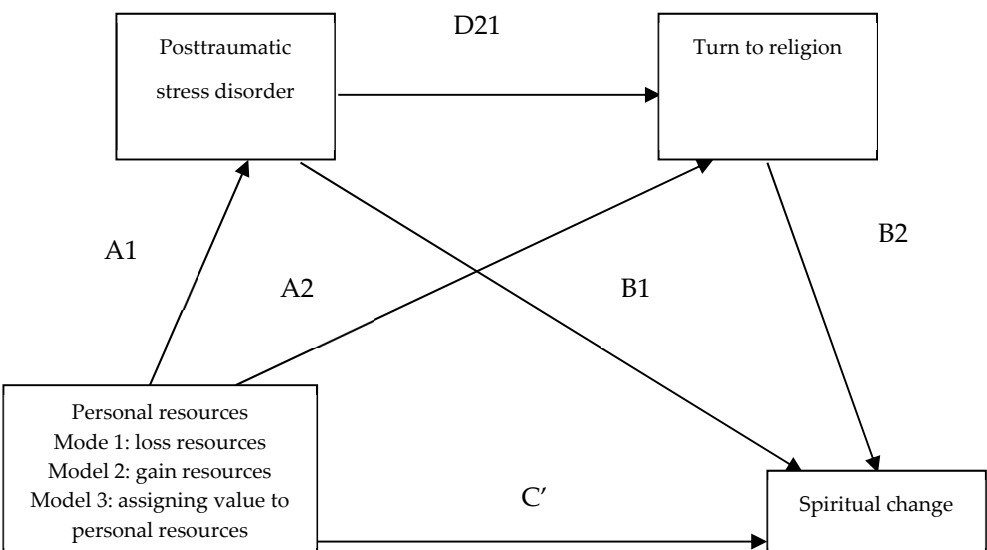

**Figure 1.** Theoretical model. Three separate mediation models were tested. The first model took into account the loss of personal resources, Model 2 took into account gain of personal resources, and Model 3 took into account the assigning value to personal resources. The indirect effect of personal resources on spiritual change by the mediating role of posttraumatic stress = A1B1; the indirect effect of personal resources on spiritual change by the mediating role of turn of religion = A2B2; the indirect effect of personal resources on spiritual change by the mediating roles of posttraumatic stress and the turn to religion = A1D21B2; direct effect of personal resources on the spiritual change while controlling for the mediators = C'.

### 3. Results

Table 1 shows the descriptive statistics and Pearson bivariate correlations for all variables.

**Table 1.** Descriptive statistics and correlations between variables.

|  | 1 | 2 | 3 | 4 | 5 | 6 |
|---|---|---|---|---|---|---|
| Resources loss [1] | - |  |  |  |  |  |
| Resources gain [2] | 0.154 ** | - |  |  |  |  |
| Assigning value to personal resources [3] | 0.021 | 0.696 *** | - |  |  |  |
| Spiritual change [4] | 0.068 | 0.006 | 0.161 ** | - |  |  |
| Posttraumatic stress disorder (PTSD) [5] | 0.184 ** | −0.043 | 0.186 ** | 0.177 ** | - |  |
| Turning to religion [6] | −0.035 | 0.214 *** | 0.326 *** | 0.352 *** | −0.107 | - |
| M (SD) | 36.56 (15.20) | 56.06 (16.98) | 66.17 (18.22) | 6.01 (2.48) | 20.51 (13.87) | 2.62 (1.75) |
| Alpha | 0.94 | 0.90 | 0.94 | 0.76 | 0.90 | 0.69 |

** $p < 0.01$, *** $p < 0.001$.

The personal resources loss had positive correlation with PTSD. The personal resources gain had positive correlation with turn to religion. The assigning value to personal resources correlated positively with spiritual change, PTSD, and turn to religion. The spiritual change had positive correlation with PTSD and turn to religion. The multicollinearity problem was not serious in model. The tolerance index was lowest at 0.45 and the variance inflation factor (VIF) was highest at 2.20.

To explain the personal resources–spiritual change link, we performed three mediation analysis in which personal resources loss, personal resources gain, and assigning value to personal resources were tested as predictors of spiritual change. PTSD and turn to religion were mediators in examined models.

The direct and indirect effects of personal resources loss on spiritual change are presented in Figure 2. The indirect effects of personal resources loss on spiritual change through PTSD and turn to religion was significant (IE = −0.001, 95% CI [−0.003; −0.000]). The loss of resources due to warfare enhances posttraumatic stress. Lower posttraumatic stress determines higher tendency to turn to religion as stress coping strategy. However, the turn to religion increases positive spiritual change. The indirect effect personal resources loss–PTSD–spiritual change was also significant (IE = 0.006, 95% CI [0.001; 0.013]). The higher level of posttraumatic stress is associated with positive spiritual change. The total indirect effect size (0.02) was small.

Figure 3 illustrates indirect effect personal resources gain–turn to religion–spiritual change was significant (IE = 0.013, 95% CI [0.005; 0.022]). The people with high level of resources tend to turn to religion in coping with stress, and tendency to turn to religion is correlated with positive spiritual change. The total indirect effect was 0.08, which suggests a small-to-moderate effect size.

Figure 4 illustrates indirect effect assigning value to personal resources–turn to religion–spiritual change was significant (IE = 0.018, 95% CI [0.010; 0.027]). The tend to turn to religion in coping with stress is positively linked with assigning value to personal resources and turn to religion correlated with positive spiritual change. The total indirect effect was 0.14, which suggests a moderate effect size.

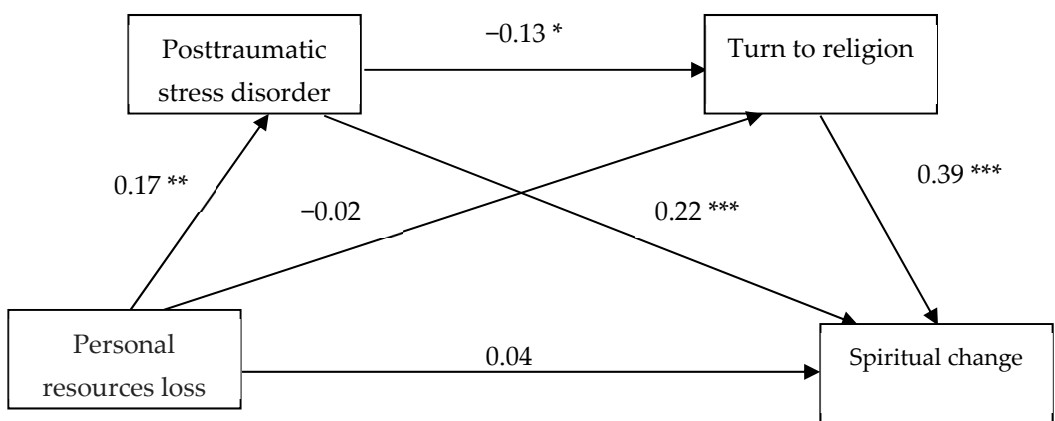

**Figure 2.** Relationships between personal resources loss and spiritual change as mediated by PTSD and turn to religion. Standardized regression coefficients for the significance level: *** $p < 0.001$; ** $p < 0.01$; * $p < 0.05$.

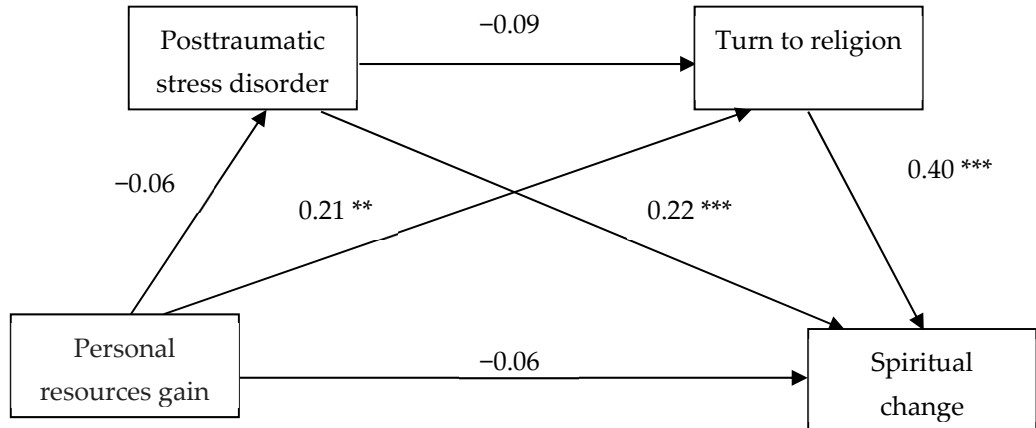

**Figure 3.** Relationships between personal resources gain and spiritual change as mediated by PTSD and turn to religion. Standardized regression coefficients for the significance level: *** $p < 0.001$; ** $p < 0.01$.

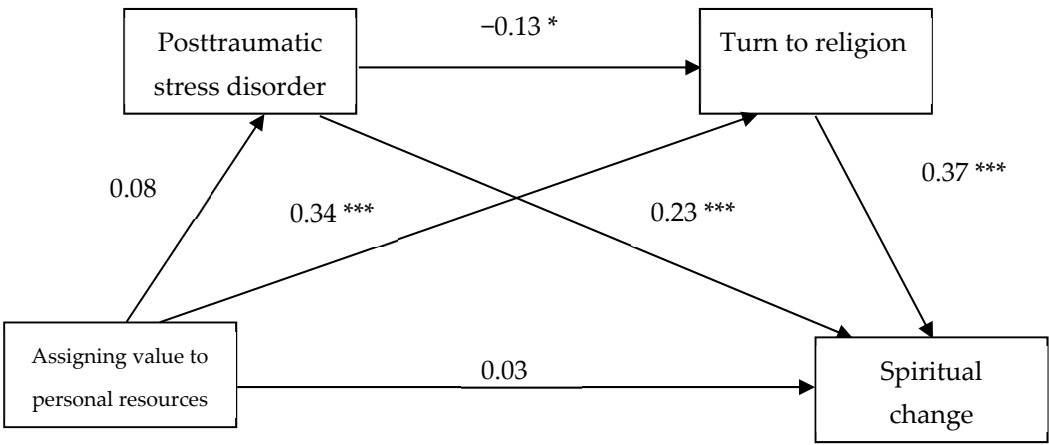

**Figure 4.** Relationships between assigning value to personal resources and spiritual change as mediated by PTSD and turn to religion. Standardized regression coefficients for the significance level: *** $p < 0.001$; * $p < 0.05$.

## 4. Discussion

The results show that there are no direct relationships between the distribution of personal resources (experienced losses, gains, and assigning a value) and posttraumatic spiritual change in the participants of military operations in Ukraine. The mediating importance of PTSD and turn to religion is noted for the existence of the analyzed relationships.

Three mechanisms could be mentioned in this context. In the first mechanism PTSD is a significant mediator between perceived personal losses and spiritual change. In the mechanism greater intensity of personal losses generates increased symptoms of PTSD, which in turn cause a higher level of spiritual change. The presented dependencies indicate that in the situation of experiencing personal resource losses, spiritual change is a consequence of PTSD, and the intensity of the increase is proportional to the perceived stress. The second mechanism is related to the fact that experienced personal losses positively correlate with PTSD, but at the same time the severity of posttraumatic stress symptoms negatively correlates with the preference for turn to religion as coping strategy, which in turn is positively associated with the manifestations of spiritual change. The presented system of dependencies suggests that in the situation of experiencing personal resource losses, positive religious coping strategies generate spiritual growth, but only when there is a slight severity of PTSD symptoms, because then they are used more often. However, in the situation with high levels of posttraumatic stress, turn to religion is used less frequently and therefore generates less spiritual change. The third mechanism is characteristic both for experienced personal gains and for assigning a value to resources. Experiencing profits in personal resources/assigning a value to personal resources is positively associated with preferring turn to religion as coping strategy, which in turn generates spiritual change. This means that the increase in experienced personal profits/assigning a value generates a more frequent preference for religious coping strategies, which in turn translates into spiritual change.

The first and the second of the mechanisms obtained confirm the basic principle of COR theory, that the loss of resources is disproportionately more important than the increase. The resource losses are felt more strongly than their profits, each subsequent loss increases the deficits of resources (de Jong 2004; Chen et al. 2015; Hobfoll et al. 2018). Due to the fact that resources tend to accumulatively merge into caravan/teams/bundles (resource caravan passageways), under condition of threat, they are lost in entire bundles/sets, e.g., in terms of personal resources there is a total loss resources relating to self-esteem, self-efficacy, optimism, and hope which are strongly correlated (Shahar et al. 2013). The fact that only losses in the analyzed dimension led to the occurrence of PTSD proves the greater strength of personal resource losses in relation to perceiving personal gains and assigning a value. The severity of posttraumatic stress symptoms is generated by the level of perceived losses and preferring a turn to religion as coping strategy only works with a low intensity of PTSD syndrome.

The third mechanism confirms other principles of COR theory. People must invest resources to protect themselves from feeling resource losses, and experienced resource gains are more valuable to a person in the situation of perceiving resource losses (Chen et al. 2015; Hobfoll et al. 2018). Thus, the feeling of personal gains, leading to a preference for positive religious coping strategies which lead to growth of the spiritual sphere, could be treated as a kind of protection against the negative effects of resource losses, and that the presented mechanism could be used more the more the person experiences loss in various dimensions of functioning. The presented regularity is consistent with the conclusions from the study results conducted within the COR theory, that a person is both less exposed to experiencing PTSD symptoms, as well as achieving mental balance faster after trauma and preferring constructive behavior when caravans of psychosocial resources are not violated (i.e., in terms of the sense of social support, optimism, meaning of life, self-efficacy, religious comfort) and/or when the person experiences profits in these areas (Thrasher et al. 2010; Zwiebach et al. 2010; Hobfoll et al. 2012; Hollifield et al. 2016; Hobfoll et al. 2020).

The presented mechanisms are also compatible with the discussion on the relationship between PTSD and spiritual change, in which arguments in support of three models of the existing relationships are presented. Firstly, PTSD and spiritual change are opposite ends of the same continuum (e.g., Frazier et al. 2001; Hagenaars and Minnen 2010). Secondly, the strength of PTSD generates spiritual change (Tedeschi and Calhoun 2004; Solomon and Dekel 2007; Dekel et al. 2012). Thirdly, spiritual change is independent of PTSD symp-

toms (Joseph and Linley 2006; Hall et al. 2008). The mechanisms obtained in the study confirm the simultaneous existence of the relationships presented in the second and third models for PTSD and spiritual change. For the mechanisms characteristic of experiencing personal losses, the second model is adequate, indicating a positive relationship between PTSD and spiritual change. However, for the feeling of personal gains, the third model is appropriate, according to which the preference for turn to religion as coping strategy leads to development in the spiritual sphere. But in this approach, changes in the spiritual sphere are not the effect of the experienced trauma (posttraumatic growth), but the effect of coping strategies undertaken as a result (Tedeschi and Calhoun 2004). The obtained mechanisms also indicate the mediating function of positive religious coping strategies for the relationship between the distribution of personal resources and positive spiritual changes. For the mechanism characteristic of the feeling of loss the mediating role is especially marked with a low level of PTSD, and with the increase in symptoms of PTSD it begins to decline. In contrast, for the mechanism characteristic of personal gains/assigning a value to personal gains, the mediation function of positive religious remedial strategies is proportional to the gains experienced (Tedeschi 1999; Harrison et al. 2001; Ryff and Singer 2002; Smith et al. 2003; Fredrickson et al. 2003). The presented relationships confirm the theory that the preference for religious coping strategies is both a factor contributing to the emergence of posttraumatic growth (when experiencing resource losses which generate PTSD) and constructive changes accompanying trauma (when experiencing PTSD) (Linley and Joseph 2004; Helgeson et al. 2006; Ogińska-Bulik 2010; Pargament et al. 2000; Tran et al. 2012; Ellison et al. 2013; Park 2013; Krok 2014; Ahles et al. 2016). Therefore, religious coping strategies could be included in the subjective characteristics of the individual, as well as mental resilience, integrated personality, internal religiosity, optimism, hope, high self-efficacy, inner sense of control, high sense of coherence, personality resilience, by which the person effectively copes with trauma (Maheux and Price 2016; Ogińska-Bulik 2016; Büssing et al. 2005; Santos et al. 2017; Areba et al. 2018; Bradshaw and Kent 2018; Pirutinsky et al. 2020).

## 5. Conclusions

In conclusion, the study aimed to investigate the mechanisms of the influence of personal resources on spiritual change. In light of our findings, we can conclude that post-traumatic stress disorder and turn to religion serve as mediators of the relationships be-tween personal resources loss on spiritual change. The loss of resources due to warfare enhances posttraumatic stress. In turn, higher posttraumatic stress determines lower tendency to turn to religion as stress coping strategy, and turn to religion correlates with spiritual change. In other words, the frequency of using this strategy is mainly visible in the low intensity of PTSD symptoms. Moreover, PTSD plays the role of mediator in a relationship between personal resources loss and spiritual change. When people experience loss of resources, they may have high levels of posttraumatic stress symptoms, and high posttraumatic stress correlated with spiritual change. The turn to religion in coping with stress plays the role of mediator in the relationship between personal resources gain/assigning a value to personal resources and spiritual change. The frequency of using this strategy is directly proportional to the intensity of experiencing personal gains or assigning a value to personal resources. The obtained results justify the postulate of conducting further research in the field of testing models considering the relationship between PTSD and spiritual change. The conducted analyses should include the assumptions of the COR theory as well as psychological, social, and situational factors that could generate posttraumatic spiritual growth. Moreover, it is worth considering the level of religiosity. In our research, we did not take into account the division into religious and non-religious people. The tested model may be modified if this division remains in the analysis. Direction and/or intensity of spiritual changes may have differed for nonreligious participants of hostilities as compared to the religious majority of the sample.

The project has been realized in a correlation scheme, as a consequence of which it is not possible to draw cause and effect conclusions based on the obtained results. The presented results are exploratory. Important limitations of the project include the lack of a detailed analysis of spiritual experiences in three time periods: before the onset of trauma, from the period of traumatic events, and posttraumatic stress symptoms, and from the time of stressors and PTSD symptoms disappearance/reduction. Another limitation of the presented results is the lack of comparison of the results explaining spiritual changes in various groups—e.g., age, gender, and the way of functioning in war circumstances (positively adjusted respondents versus individuals exhibiting adaptation problems). For example, the results of the research show that the presence of a significantly higher level of spiritual change is noted in women, compared to men, or in the elderly compared to younger people (Tedeschi and Calhoun 1996; Ogińska-Bulik and Juczyński 2010). Further research should answer the question whether the models presented in the article, which explain posttraumatic spiritual changes, are universal in nature, or whether there are mechanisms specific to different populations in the analyzed context. An additional limitation of the research are the results relating to the mediating importance of the religious coping strategy for the relationship between the personal resources and spiritual change. In future research, it is worth comparing the indirect significance of positive and negative religious strategies of coping with trauma for the occurrence of spiritual change (Desai and Pargament 2015).

**Author Contributions:** Conceptualization, I.N.; methodology, I.N., J.C. and K.J.; formal analysis, K.J.; investigation, I.N.; data curation, K.J.; writing—original draft preparation, I.N. and K.J.; writing—review and editing, I.N., K.J., J.C., M.K.-P. and P.W.; visualization, K.J.; project administration, I.N. All authors have read and agreed to the published version of the manuscript.

**Funding:** This research received no external funding.

**Institutional Review Board Statement:** The study was conducted according to the guidelines of the Declaration of Helsinki, and approved by the Ethics Committee of the Institute of Sociological Sciences of the John Paul II Catholic University of Lublin (protocol code: KEB-IS-3/2019).

**Informed Consent Statement:** Informed consent was obtained from all subjects involved in the study.

**Data Availability Statement:** The data presented in this study are available on request from the corresponding author.

**Conflicts of Interest:** The authors declare no conflict of interest.

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
