# Peer review of "Personal Resources and Spiritual Change among Participants’ Hostilities in Ukraine: The Mediating Role of Posttraumatic Stress Disorder and Turn to Religion"

_religions, doi:10.3390/rel12030182_

Round 1

Reviewer 1 Report

I appreciate the author's thorough review of the literature to supply the rationale for the study. I have just a few minor suggestions to the article.

There are several very minor mistakes that need correction in editing, including repetition of ICD name, Groth misspelled in reference list, abbreviation of including, misspelling of analysis, Participants, correlates, DSM-III-R needs a hyphen, etc.  Donbas/Donbass seems to be spelled differently in the Participants section.

I’m uncertain as to the use of the term “unambiguous” in the opening paragraph, especially in light of the very wide range of prevalence rates found there.  The Bogic article pointed out that the wide variance in prevalence numbers are due to methodological problems in studies, with the lower prevalence rates usually being those with the more rigorous controls.  At least some statement needs to be added to note that prevalence rates, while varying widely, suggest that the problem is substantial.  And I think the phrase “no unambiguous…” should be deleted given these varied numbers and the remaining ambiguity in prevalence rates in studies of PTSD.  Maybe convey that while the data is ambiguous the severity of the problem is unambiguous.

Regarding references in text, it is a bit less confusing if sentences are referenced earlier rather than paragraphical information at the end.  For example, in the discussion about spiritual change on p. 2 it is rather late in the list of declarative sentences to give the references at the end. 

I didn’t find any problems with the statistics, results, or conclusions made.

Author Response

Dear Sir or Madam,

We would like to thank you for taking the time and effort necessary to review the manuscript. We sincerely thank for the constructive comments and suggestions, which helped us to substantially improve our manuscript. Below we present our responses to the suggestions and recommendations.

Yours faithfully
(removed for peer review)

Response to Reviewer Comments

Point 1: There are several very minor mistakes that need correction in editing, including repetition of ICD name, Groth misspelled in reference list, abbreviation of including, misspelling of analysis, Participants, correlates, DSM-III-R needs a hyphen, etc.  Donbas/Donbass seems to be spelled differently in the Participants section.

Response 1:  Thank you for pointing out the mistakes in the text, we have made the necessary spelling corrections in the article /lines: 32-33; 226; 241; 268; 286; 319; 349; 481/, we used the abbreviation PTSD in the whole manuscript, we have corrected misspell (Groth) in reference list /line 563/.

Point 2: I’m uncertain as to the use of the term “unambiguous” in the opening paragraph, especially in light of the very wide range of prevalence rates found there.  The Bogic article pointed out that the wide variance in prevalence numbers are due to methodological problems in studies, with the lower prevalence rates usually being those with the more rigorous controls.  At least some statement needs to be added to note that prevalence rates, while varying widely, suggest that the problem is substantial.  And I think the phrase “no unambiguous…” should be deleted given these varied numbers and the remaining ambiguity in prevalence rates in studies of PTSD.  Maybe convey that while the data is ambiguous the severity of the problem is unambiguous.

Response 2:  Thank you for your comment, we agree with the Reviewer that the statement raises a doubtful opinion. According to Reviewer’s suggestion, we corrected the sentence which introduces the results of our study /lines 41-42: While the severity of the problem of PTSD in people who participate in hostilities is unambiguous, the results of research are inconsistent/.

Point 3: Regarding references in text, it is a bit less confusing if sentences are referenced earlier rather than paragraphical information at the end.  For example, in the discussion about spiritual change on p. 2 it is rather late in the list of declarative sentences to give the references at the end.

Response 3: We agree with the Reviewer's suggestion, we have made the necessary corrections, both in the introduction /lines 204; 207-209; 211-213 / as and in discussions / lines: 462-463; 467-468; 472-474/.

Reviewer 2 Report

The manuscript explores COR theory in the context of spiritual changes associated with the experience of trauma, specifically hostilities in Ukraine. I very much appreciate scholarship that attends religion/spirituality as it pertains to trauma recovery. It is also commendable that the study explores both psychological distress (e.g., PTSD symptoms) and post traumatic growth, as it acknowledges challenges as well as strengths that accompany traumatic events. Below you will find feedback and recommendations for improvement of the manuscript: 

  1. I would better define "participated in hostilities". As you discuss literature related to military and civilians, be clear about what constitutes a hostility.
  2. In the literature review, I began to wonder about the effect of multiple and/or chronic trauma(s)? In other words, how might other trauma associated with hostilities or independent of them influence symptoms of PTSD? For example, people build resilience over time or find one specific trauma more salient than others. You might make mention of this early and note how you assessed for it, if at all, in your method/analyses. 
  3. I would distinguish between religion and spirituality as constructs. They seem to be used somewhat interchangeably, but spirituality may or may not be associated with belief in a deity. Which did you mean to explore? If both, how did you address this difference? 
  4. Some literature that you may find useful: There are a series of articles related to mortality salience and religious/spiritual belief change (e.g., Heflick &. Goldenberg, Wansink & Wansink, Jong et al.). Though this is slightly more focused than the range of traumas related to hostilities, it challenges the idea that life stressors lead to increases in religion/spirituality. You might also look at Galen & Kloet (2011) who suggest it is the strength of certainty in one's beliefs (whether religious or nonreligious) that is more predictive of well-being than religious belief in and of itself. 
  5. In line with these suggestions, the paper seems to almost assume a level of spirituality and religiosity among potential participants. What about nonreligous/nonspiritual people? What do we know about them? How are they accounted for in your method?
  6. With regard to hypotheses, it would be helpful to define terms like spiritual change and turn to religion before you state them. What do you mean by that? In what direction to you hypothesize these variables will change/turn?
  7. Minor points: There is a typo in "participants" in heading 2.1. You might also say "living with a partner" or similar to avoid the assumption that participants lived with other participants (as it's currently written). 
  8. I'd like to see more demographic data, if possible (e.g., sexual orientation, race/ethnicity, social class or socioeconomic status). It seems particularly important to note the religious or spiritual orientation of the sample, if known, given your variables of interest. 
  9. In the Instruments, some of the descriptions of measures need example items and/or reliabilities. These are used/reported inconsistently in this section. I would also encourage specific reliabilities (past and in the current study) rather than simply stating "satisfactory". 
  10. Why not use the PCL-5 (the most updated version of the measure)? 
  11. Given your use of the spiritual change subscale only, include example items.
  12. Was the turn to religion measure a 1-item measure. Given 1-item measures have limitations, please address. 
  13. I would avoid saying "mediation was proved" and rather just speak to the significance of indirect effects. 
  14. Your figure depicts a more complex model but you say you performed three independent mediation analyses. I understand the figure, but am afraid it inadvertently suggests that all of these relationships were tested as one model. Please clarify. 
  15. When discussing the relationships, please address effect size (e.g., "a small, significant relationship with X, moderate relationship with Y, etc."). 
  16. Although you note a limitation at the end of the manuscript, a more comprehensive section of limitations is necessary. 

Author Response

Dear Sir or Madam,

We would like to thank you for taking the time and effort necessary to review the manuscript. They allowed us to look at the discussed problems from a wider perspective and are also valuable in relation to further research that we are conducting. The constructive comments and suggestions helped us to substantially improve our manuscript.  Below we present our responses to the suggestions and recommendations (additionally, we append detailed point-by-point responses in the document uploaded as an attachment).

Yours faithfully
(removed for peer review)

Responses to Reviewer Comments

Point 1: I would better define "participated in hostilities". As you discuss literature related to military and civilians, be clear about what constitutes a hostility.

Response 1: We agree with the Reviewer that the text does not explain how we understand - in the context of the analyzes being conducted - warfare. In the introduction, we have completed the information which is based on the relevant literature; lines 33-42.

Point 2: In the literature review, I began to wonder about the effect of multiple and/or chronic trauma(s)? In other words, how might other trauma associated with hostilities or independent of them influence symptoms of PTSD? For example, people build resilience over time or find one specific trauma more salient than others. You might make mention of this early and note how you assessed for it, if at all, in your method/analyses.

Response 2: We strongly agree with the Reviewer’s opinion that the literature on the subject also does not contain an uniform position regarding the relationships between the size of the trauma and the severity of posttraumatic transformations. In the context of the research, some authors indicate that the size of post-traumatic changes depends on the intensity of the trauma, i.e. the more intense the traumatic experience, the greater the chance of positive changes resulting from this type of experience. However, other authors claim that this is a curvilinear relationship, i.e. that moderate trauma promotes transformation in the way of perceiving circumstances, while low or extremely high intensity events do not lead to such changes. We supplemented the text with reference to the Reviewer's recommendation. Moreover, we would like to mention that we are conducting the research in the field of identifying predictors of the intensity of PTSD for spiritual change, i.e. we are interested in - as the Reviewer notices - whether other experiences related to warfare may affect PTSD symptoms, and at the same time whether the level of PTSD determines the spiritual change and in what direction (e.g., strong PTSD causes less spiritual change). The results of this research may bring a new quality to the understanding of the mechanisms which we are analyzing; lines 65-73.

Point 3: I would distinguish between religion and spirituality as constructs. They seem to be used somewhat interchangeably, but spirituality may or may not be associated with belief in a deity. Which did you mean to explore? If both, how did you address this difference? 

Response 3: We agree with the Reviewer’s opinion that spirituality does not have to be related to faith in God. In the literature, there is a dispute on how to define spirituality and religiosity, and what kind of relations exist between these concepts (Oman & Thoresen 2003). The authors of the concept of post-traumatic growth (PTG), which includes changes of a spiritual nature (the domain of spiritual change), adopted a broad concept of spiritual growth, which includes both a religious component and elements beyond the religious sphere: The domain consists of two items: “A better understanding of spiritual matters” and “I have a stronger religious faith”. The authors emphasizes that spiritual changes taking place as part of posttraumatic growth are not the same as religious changes. But a structured religious framework can facilitate the occurrence of PTG, which may provide a stronger sense of meaning, increase of social support, acceptance of suffering, and a change of belief system that can be integrated into the individual's life (Overcash et. Al. 1996; Silberman, 2005; Hussain & Bhushan, 2011; Castella & Simmonds 2013). According to another definition of spirituality, this is the searching for holiness in God, in divine beings, or in a transcendent reality. Religion, on the other hand, provides the social and institutional context for this kind of exploration (Pargament et al. 2006). Consequently, we assumed that the strategy of coping with stress, consisting in turning to religion, may be a moderator in the relationship between personal resources and spiritual change.

Point 4: Some literature that you may find useful: There are a series of articles related to mortality salience and religious/spiritual belief change (e.g., Heflick &. Goldenberg, Wansink & Wansink, Jong et al.). Though this is slightly more focused than the range of traumas related to hostilities, it challenges the idea that life stressors lead to increases in religion/spirituality. You might also look at Galen & Kloet (2011) who suggest it is the strength of certainty in one's beliefs (whether religious or nonreligious) that is more predictive of well-being than religious belief in and of itself. 

Response 4: We have added to our article the conclusions from the research which were indicated by the Reviewer (Galen and Kloet 2011). We assume that spiritual change in the context of experienced trauma concerns / refers to a better understanding of spiritual problems and / or a deepening of the spiritual experience. The presented understanding of changes in the spiritual sphere means that before the occurrence of the trauma, the person had spiritual experiences (in such circumstances, traumatic events may lead to their intensification or decline), or one of the effects of the trauma is a turn towards triggering spiritual experiences. The change in the spiritual dimension is the result of the transformation taking place in the way of perceiving the events that took place by processing the information contained in the cognitive scripts about the experienced trauma in the context of searching for the meaning of the event and / or the emergence of reflection on its importance for further functioning. Although some people report loss of faith and an increase in existential despair after experiencing trauma, evidence suggests that some people after trauma experience changes as increasing their sense of purpose and purpose in life, deepening spirituality and strengthening faith (Tedeschi and Calhoun, 2004) lines 172-175.

Point 5: In line with these suggestions, the paper seems to almost assume a level of spirituality and religiosity among potential participants. What about nonreligous/nonspiritual people? What do we know about them? How are they accounted for in your method?

Response 5: We appreciate your opinion. At the same time, we would like to inform that we are working on the indicated theoretical model. One of the areas of the analyzes is the identification of variables (moderators) that might affect the described dependencies, e.g. the issue of religion, the intensity of religiosity of participants in military operations. In the mechanism that we presented in the study, we assume that traumatic events lead to the intensification (or decrease) of spiritual change, or one of the effects of trauma is a shift towards triggering spiritual change (regardless of the initial level of religiosity). For this reason, in this research project, we have not distinguished between religious / non-religious respondents. We agree with the Reviewer that further research should answer the question whether the models presented in the article, which explain posttraumatic spiritual changes, are universal in nature, or whether there are mechanisms specific to different populations in the analyzed context.

Point 6: With regard to hypotheses, it would be helpful to define terms like spiritual change and turn to religion before you state them. What do you mean by that? In what direction to you hypothesize these variables will change/turn?

Response 6: We have made an appropriate explanation in the text, thanks to which the article - we hope - is more understandable; lines 74-91. At the same time, we would like to emphasize that in our research we assumed that personal resources correlate with the mediator which is PTSD, while the intensity of trauma triggers a strategy of coping with stress based on turning to religion. We wanted to confirm that these variables, ie PTSD and the turn to religion, mediate the relationship between resources and spiritual change. Due to the ambiguity of the described relationships (e.g. between mediators, initially we assumed that higher PTSD triggers a strategy of turning to religion, but this correlation is negative) in the literature we did not assume the direction in the hypothesis.

Point 7: Minor points: There is a typo in "participants" in heading 2.1. You might also say "living with a partner" or similar to avoid the assumption that participants lived with other participants (as it's currently written). 

Response 7: We have corrected typo in the line 226. We strongly agree that the statement we used was incorrect, we used the phrase suggested by the Reviewer; line 229.

Point 8: I'd like to see more demographic data, if possible (e.g., sexual orientation, race/ethnicity, social class or socioeconomic status). It seems particularly important to note the religious or spiritual orientation of the sample, if known, given your variables of interest. 

Response 8: We agree with the Reviewer that additional variables may constitute an important context in the perspective of the analyzed phenomena. In the characteristics of the respondents, we have added the information we have at our disposal, i.e. on the material situation of the respondents, material status, education, confession; lines 229-234.

Point 9: In the Instruments, some of the descriptions of measures need example items and/or reliabilities. These are used/reported inconsistently in this section. I would also encourage specific reliabilities (past and in the current study) rather than simply stating "satisfactory". 

Response 9: We have added sample items from COR evaluation and PCL-C and items which make up the analyzed subscales, i.e. turn to religion (MINI-COPE), lines 296-297; and spiritual change (PTG) lines 308-309. We supplemented the descriptions with the measures of reliability indicated by the authors of the tools which were used in the study; lines 272; 294-295 and 304-305.

Point 10: Why not use the PCL-5 (the most updated version of the measure)? 

Response 10: Although there is a PCL-5 version that meets the DSM-V criteria, we used PCL-C. This tool - thanks to very good psychometric properties - is used in many current studies For example: Beatriz Ponce de León, et. al. 2018; Ya-Xi Wan et. al., 2020; Robert Hatch et. al., 2020, Jeffrey M Osgood et. al., 2019; Clark and Walker 2020). The use of PCL-C has its substantive and practical justification. We used this method in a pilot study in 2015. Moreover, the method has been translated into Ukrainian by 3 independent translators. We were sure that it was understood by the respondents. Moreover, we would like to mention that the civil version of the PCL was used to enable respondents to report symptoms of increased stress associated with any traumatic events, and not only symptoms caused by war experiences. We assumed that the assessment of posttraumatic stress intensification symptoms, taking into account military and non-military sources of trauma, is important when considering the general mental health of participants in military operations. We have supplemented our text with this information; lines 270-277.

Point 11: Given your use of the spiritual change subscale only, include example items.

Response 11: We agree with the opinion. In the description of the scale, we added the items that make up the subscale; lines 257-258; 260-264; 277-281; 296-297; 308-309.

Point 12: Was the turn to religion measure a 1-item measure. Given 1-item measures have limitations, please address. 

Response 12: We used the Turn to Religion subscale as one of the strategies identified using the MINI-COPE tool (Polish adaptation of Brief COPE). There are two items in the strategy. We indicated the content of the item in the description of the tool.

Point 13: I would avoid saying "mediation was proved" and rather just speak to the significance of indirect effects. 

Response 13: We agree that the statement "mediation was proved" needs to be changed. We have corrected the text (If this CI does not include zero, the analyst concludes that there is statistically significant mediation); lines 321-325.

Point 14: Your figure depicts a more complex model but you say you performed three independent mediation analyses. I understand the figure, but am afraid it inadvertently suggests that all of these relationships were tested as one model. Please clarify. 

Response 14: We have made a more detailed description and correction in the figure which presents the assumptions of our model; lines 327-329.

Point 15: When discussing the relationships, please address effect size (e.g., "a small, significant relationship with X, moderate relationship with Y, etc."). 

Response 15: We have made additions to the article and to the description of the statistical methods which were used. We added the information about the assumptions made in the assessment of the size of the effects, and we have added the literature; lines 360; 370-371; 380-381.

Point 16: Although you note a limitation at the end of the manuscript, a more comprehensive section of limitations is necessary. 

Response 16: We have discussed the limitations of research in the last paragraph of the article. We pointed out, among others the limitations of correlation studies and substantive limitations, e.g. the lack of a detailed analysis of spiritual and religious experiences in three time periods: before the onset of trauma, from the period of traumatic events and experiencing symptoms of posttraumatic stress, and from the time when stressors disappear and PTSD symptoms disappear / decrease. This section also summarizes the postulates for future research in this field; lines 496-513.

Round 2

Reviewer 2 Report

I appreciate your attention to previous comments. The revisions you made helped me, as a reader, better understand the constructs you explored and their relationship to the literature you review. In particular, the transparency and definitions you provided in the literature review were helpful, as was your explanation of the use of the PCL-C. 

The expansion of your discussion of demographics was also appreciated. Some questions remain: what do you mean by "higher education qualification"? You note that "most" reported a bad financial situation, but the percentage is only 46.5% (fewer than half). You also note that most are Orthodox, but, particularly given your variables of interest, who comprised the remainder of the sample. Were they also religious? How many, if any, were nonreligious? 

Discomfort remains for me regarding the presumption of religiosity among participants, in general - especially given the language of the items to assess spiritual change. Your explanation of your choices states that you explored these variables, including spiritual change, regardless of (non)religious orientation pre-trauma. I hear you and there is literature that suggests this assumption of increased religiousness/spirituality due to trauma/mortality salience is inaccurate (e.g., Heflick & Goldneberg, Wansink & Wansink, Jong et al - the "atheists in foxholes" series). 

I think there could be more integration of this literature (as you did with Galen & Kloet) or you could be more transparent about the exclusion of nonreligious people and/or omission of related literature - that it wasn't the focus of the study but you acknowledge this omission. Perhaps adding this to your limitations would accomplish the task. For example, if you included nonreligious people in the analysis (that's unclear from the current Participants section), that may have influenced your findings. Or, group differences were not explored and may have differed for nonreligious participants as compared to the religious majority of the sample. 

To be clear, I don't think it needs a major change/restructuring due to this omission - just an acknowledgment in a place or two. 

Small errors: line 228 "lived with..."; typo on p. 4 (Galen)

Author Response

Dear Sir or Madam,

We would like to thank you for taking the time and effort necessary to review the new version of manuscript. We sincerely thank for the constructive comments and suggestions. Below we present our responses.

Point 1: Some questions remain: what do you mean by "higher education qualification"? You note that "most" reported a bad financial situation, but the percentage is only 46.5% (fewer than half). You also note that most are Orthodox, but, particularly given your variables of interest, who comprised the remainder of the sample. Were they also religious? How many, if any, were nonreligious? 

Response 1: Thank you for your opinion. We agree that term ,,majority” is incorrect. In the new version of the manuscript, we used the phrase "the highest number", both in the description of the education and the financial situation of the respondents (lines 235; 237). By "higher education qualification" we meant "higher education" (line 236) [including Initial level (short cycle) of higher education]. We have removed the word “qualification.” We have also added the information about the religiosity of our respondents (lines 239-240).

Point 2: Discomfort remains for me regarding the presumption of religiosity among participants, in general - especially given the language of the items to assess spiritual change. Your explanation of your choices states that you explored these variables, including spiritual change, regardless of (non)religious orientation pre-trauma. I hear you and there is literature that suggests this assumption of increased religiousness/spirituality due to trauma/mortality salience is inaccurate (e.g., Heflick & Goldneberg, Wansink & Wansink, Jong et al - the "atheists in foxholes" series). I think there could be more integration of this literature (as you did with Galen & Kloet) or you could be more transparent about the exclusion of nonreligious people and/or omission of related literature - that it wasn't the focus of the study but you acknowledge this omission. Perhaps adding this to your limitations would accomplish the task. For example, if you included nonreligious people in the analysis (that's unclear from the current Participants section), that may have influenced your findings. Or, group differences were not explored and may have differed for nonreligious participants as compared to the religious majority of the sample. To be clear, I don't think it needs a major change/restructuring due to this omission - just an acknowledgment in a place or two.

Response 2: We are very grateful for showing us a broader perspective in further research on the spiritual changes resulting from the trauma. We agree that the division into religious and non-religious respondents may bring interesting solutions and modifications to our model. For the purposes of the manuscript, in the participants section, we have added the information about religiosity; we have emphasized that not all war experiences lead to focus on a relationship with God lines: 209-212; and in the limitations of the research we have noted this problem / limitation of our research, i.e. no division into religious and non-religious respondents; 500-504.

Point 3: Small errors: line 228 "lived with..."; typo on p. 4 (Galen)

Response 3: We have made the necessary corrections: lines 233 and 173